# LPS induces inflammatory chemokines via TLR-4 signalling and enhances the Warburg Effect in THP-1 cells

**Philemon Ubanako[1], Ntombikayise Xelwa[1], Monde Ntwasa[ID][2]\***

**1** School of Molecular & Cell Biology, University of the Witwatersrand, Johannesburg, Republic of South Africa, **2** Department of Life & Consumer Sciences, University of South Africa, Florida, Johannesburg, Republic of South Africa

\* ntwasmm@unisa.ac.za

## Abstract

The Warburg Effect has emerged as a potential drug target because, in some cancer cell lines, it is sufficient to subvert it in order to kill cancer cells. It has also been shown that the Warburg Effect occurs in innate immune cells upon infection. Innate immune cells play critical roles in the tumour microenvironment but the Warburg Effect is not fully understood in monocytes. Furthermore, it is important to understand the impact of infections on key players in the tumour microenvironment because inflammatory conditions often precede carcinogenesis and mutated oncogenes induce inflammation. We investigated the metabolic programme in the acute monocytic leukaemia cell line, THP-1 in the presence and absence of lipopolysaccharide, mimicking bacterial infections. We found that stimulation of THP-1 cells by LPS induces a subset of pro-inflammatory chemokines and enhances the Warburg Effect. Surprisingly, perturbation of the Warburg Effect in these cells does not lead to cell death in contrast to what was observed in non-myeloid cancer cell lines in a previous study. These findings indicate that the Warburg Effect and inflammation are activated by bacterial lipopolysaccharide and may have a profound influence on the microenvironment.

## Introduction

Cancer cells uncouple glycolysis and oxidative phosphorylation (OXPHOS), a phenomenon known as the Warburg Effect after the German Scientist Otto Warburg who first described it [1]. This metabolic reprogramming is one of the ten hallmarks of cancer and has been found to occur in activated immune cells as well [2–4]. Previously, we have shown that the reversal of the Warburg Effect in some cancer cell lines by the introduction of exogenous pyruvate is sufficient to kill them. In contrast, this strategy promotes survival in a normal cell line even when exogenous pyruvate is combined with a genotoxic agent suggesting that exogenous pyruvate can be used as an adjunctive to protect normal cells during chemotherapy [5].

Evidence shows that temporal metabolic shifts occur in immune cells. For example, it is known that myeloid cells, which have multiple functions such as killing, migration and toxin production, utilize Warburg metabolism for ATP production [6,7]. In contrast, lymphocytes

**Funding:** The National Research Foundation (South Africa) [GUN: 105567] supported this research

awarded to MN. PU conducted the bulk of the experiments analysis of results. NX conducted the metabolic studies. MN conceived the project and analysed the data. All authors participated in drafting the manuscript.

**Competing interests:** The authors have declared that no competing interests exist.

utilize mainly OXPHOS to produce ATP via glutaminolysis, β-oxidation of fatty acids as well as via glycolysis [8]. Metabolic shifts have also been shown to be associated with the differentiation of some immune cells. For example, $T_H1$ cells were shown to differentiate to $T_{reg}$ cells upon sabotage of the glycolytic phenotype by treatment with the glucose analogue, 2-deoxyglucose [9].

Myeloid cells are often activated by exposure to environmental stimuli, resulting in metabolic shifts. For example, bacterial lipopolysaccharide (LPS) induces metabolic reprogramming in macrophages and dendritic cells via Toll-like Receptor 4 (TLR4) by at least four pathways. These include generating inducible nitric oxide synthase (iNOS) and subsequent suppression of OXPHOS, activation of mammalian target of rapamycin (mTOR), increase in glucose uptake and glycolysis, upregulating key glycolysis enzymes and inhibiting the AMP-activated protein kinase (AMPK) pathway [10,11]. Although, the metabolic reprogramming that occurs in macrophages and in dendritic cells has been studied extensively [10], little is known about metabolic shifts in monocytes that are exposed to infections especially the impact of bacterial infection on their metabolic programme. Indeed, the impact of infection on cancer cells is poorly understood, even though malignancies such as colorectal cancer are constantly exposed to pathogens. In these environments, innate immune cells associated with epithelial linings play critical roles as the first line of defence. It is thus necessary to investigate the interactions between pathogens and cancerous cells.

THP-1 is a human monocyte leukaemia cell line, which differentiates into macrophage-like cells when treated with phorbol esters. When differentiated, THP-1 cells behave like native monocyte-derived macrophages. They are activated by LPS via TLR-4 and activation of NF–κB, the key transcription factor that activates expression of effector genes. Optimal response to LPS occurs when the CD14 cell surface protein is also present and is characterized by the production of the pro-inflammatory interleukin-8 (IL-8) [12]. In THP-1 cells, LPS has been shown to induce the production of pro-inflammatory cytokines such as interleukin-1β (IL-1β), tumour necrosis factor (TNF-α) and interleukin-6 (IL-6) via the TLR4- NFκB signaling pathway [12,13]. In primary monocytes in has been shown that LPS induces a metabolic shift similar to the Warburg Effect in the presence of glucose but rely on mitochondrial respiration in the absence of glucose or abrogation of the Warburg Effect [14]. Terminally differentiated monocytes such as dendritic cells and macrophages are known to exhibit the Warburg effect in response to pathogen invasion [10]. About 20% of all cancers are associated with infection [15]. The microbial oncogenic transformation has been shown to be elicited by chronic inflammation as a result of prolonged infection [16,17]. Nonetheless, the mechanisms that drive infection-associated carcinogenesis are not fully understood and the precise molecular events occurring in the tumour microenvironment are still not fully understood. Therefore, the THP-1 cell line provides an attractive model by which to study metabolic reprogramming in human monocytes especially in circumstances where oncogenic transformation and infection occur simultaneously. A cocktail of pro-inflammatory cytokines produced in the tumour microenvironment has been shown to influence carcinogenesis by enhancing proliferation, invasion, chemoresistance and resistance to apoptosis [18–20]. Thus, it appears that metabolic reprogramming in macrophages and dendritic cells is associated with immune activation and not with cell proliferation [10,11,21]. Response to similar stress is not clear in monocytes.

In differentiated THP-1 cells, metabolic shifts may be driven by hypoxia-inducible factor 1α (HIF-1α). It has been demonstrated that when activated by phorbol myristate HIF-1α and HIF-1β are upregulated in THP-1 cells as well as in peripheral blood monocytes. This is blocked by inhibitors of P13K, tyrosine kinases, MEK, mTOR, and protein kinase C (PKC) [22]. In contrast, LPS-induced HIF-1 expression is dependent on TLR4 and reactive oxygen species (ROS) as it is abrogated by the anti-oxidants *N*-acetylcysteine and thioredoxin.

However, LPS-induced HIF-Iα expression in differentiated THP-1 cells was found to be independent of NFκB as it was not inhibited by reservatol and BAY11-7802 [23]. The key observation by [23] was that LPS induces activation of HIF-1α in differentiated THP-1 cells but not in undifferentiated cells suggesting that monocytes and macrophages respond differently to LPS activation. In this study, we investigate the impact of infection on the metabolic programme of THP-1 cells, which represent transformed undifferentiated myeloid cells. Using the THP-1 cell line as a model, we also investigate the impact of bacterial infection on cancer cells that should already have a glycolytic phenotype.

Our findings indicate that LPS perturbs the cell cycle and enhances the Warburg Effect in the THP-1 cells. Moreover, the Warburg Effect does not appear to be associated with cell proliferation as observed in most cancer cells.

## Materials and methods

### Cell lines and culture conditions

The Acute monocytic leukaemia cell line (THP-1 ATCC® TIB-202™) was obtained from American Type Culture Collection (ATCC), Virginia, USA. THP-1 cells were cultured in Roswell Park Memorial Institute (RPMI) 1640 Medium (Sigma-51536C), supplemented with 10% Foetal bovine serum (FBS) (Biowest, S181H-500) and 1% Penicillin/Streptomycin (Sigma, P4333) at 37˚C, with 5% $CO_2$ and 95% relative humidity.

### Cell stimulation and treatments

Bacterial lipopolysaccharides (LPS) from *Escherichia coli* (TLR4 agonist) (Sigma, O111:B4) was dissolved in 1X phosphate- buffered saline (PBS). Polymyxin B sulphate (TLR4 antagonist) (Sigma-81334) was dissolved in water. The LPS-polymyxin B combination was pre-incubated at 37˚C for 2 hours before treating cells. Methyl pyruvate (Sigma-371173) was dissolved in 1XPBS and used at 8.8 mM final concentration. Carbonyl cyanide-p-trifluoromethoxyphenyl-hydrazone (FCCP) (Sigma-C2920) was dissolved in 95% ethanol at 10 μM and used as a positive control for the mitochondrial membrane potential assay.

### Reverse transcription polymerase chain reaction

Following treatment with 5, 10, and 20 ng/ml of LPS for 24 hours, RNA was isolated using the Trizol method as described in manufacturer's brochure and RNA concentration was determined using the Nanodrop spectrophotometer (Thermofischer Scientific, CA, USA). Complementary DNA (cDNA) was synthesized from 1 μg of total RNA, using the Revert Aid first strand cDNA synthesis kit (Thermofischer Scientific, K1622) and oligo (dT) primers. The reaction was run in a PCR thermal cycler at 42˚C for 1 hour.

TLR4 was amplified by PCR in a 25 μL reaction volume containing 200 nM forward and reverse primers, 1.5 μl of cDNA, 12.5 μL of 2X PCR mastermix (New England Biolabs, M0270) and 10 μl of nuclease-free water. Thermal parameters were set as follows: Initial denaturation at 94˚C for 30 seconds, followed by a 30-times cycle of 94˚C (30 seconds), 60˚C (60 seconds), and 68˚C (30 seconds) followed by one cycle at 68˚C for 5 minutes. To further validate ELISA array cytokine screening, gene-specific primers for upregulated cytokines were designed (NCBI Primer Blast, NCBI) and synthesized by (Inqaba Biotec, Pretoria, South Africa) Table 1 following the same procedure described earlier.

The PCR products were visualized on 1% agarose gel containing 0.5 μg/ml of ethidium bromide by imaging using the ChemiDoc™ MP Imager (Biorad, CA, USA). Densitometric analysis

**Table 1. PCR primers used.**

| Gene | Forward primer | Reverse primer | Product size |
|---|---|---|---|
| IL-8 | AAGGTGCAGTTTTGCCAAGG | CAACCCTCTGCACCCAGTTT | 204 |
| RANTES(CCL5) | CTGCCTCCCCATATTCCTCG | TCGGGTGACAAAGACGACTG | 137 |
| MIP-1β (CCL4) | GCACCAATGGGCTCAGAC | GCTCAGTTCAGTTCCAGGTC | 211 |
| MDC (CCL22) | GCGTGGTGTTGCTAACCTTC | AGGGCCAGGGGCATCTAAT | 649 |
| RPL01 | GGCAAGAACACCATGATGCG | TCGAACACCTGCTGGATGAC | 416 |
| GAPDH | GAAGGTGAAGGTCGGAGTC | GAAGATGGTGATGGGATTTC | 226 |
| TLR4 | CCAGTGAGGATGATGCCAGAAT | GCCATGGCTGGGATCAGAGT | 67 |

was performed on the bands using MyImageAnalysis™ software (Thermofischer Scientific, MA, USA). The bands were normalized against GAPDH and RPlO1 as reference genes.

## Cytokine expression analysis using ELISA array

The Human, bacterial Toll-like receptor Multi analyte ELISA array kit ((MEH-008A), Hilden, Germany) was used to assess a cytokine profile in response to LPS stimulation as a screening exercise. The wash buffer, bovine serum albumin (BSA), sample dilution buffer, avidin-HRP conjugate and antigen standard cocktail were prepared according to the manufacturer's instructions.

The 12 wells in each row of the 96-well ELISA plate (Multi-analyte ELISA array kit, MEH-008A) are pre-coated with monoclonal antibodies to TNF-α, IL1β, IL6, IL12, IL17A, IL8, MCP-1, RANTES, MIP-1α, MIP-1β, MDC and eotaxin. Into sample wells, 50 μl each of assay buffer THP-1 extracts were added. Using a multi-channel pipette, 50 μl of sample dilution buffer was added to all the negative control wells. To prepare the positive control,50 μl of the final antigen standard cocktail was added. The reactions were then processed as described in the manufacturer's manual.

Absorbance was read using a microplate spectrophotometer (Multiskan™ GO, Thermofischer Scientific, MA, USA) at 450 nm and 570 nm. The readings at 570 nm were subtracted from those at 450 nm for wavelength correction. For each antigen, the corrected absorbance values were calculated by subtracting the absorbance of the negative control from the absorbance of test samples and the results were plotted on a bar graph.

## Glycolysis/Oxygen consumption dual assay

Oxygen consumption was assessed using MitoXpress® Xtra (Cayman Chemicals- 600801) after the cells were treated either with LPS or with a combination of LPS and Polymyxin B for 24 hours. Cells were seeded in 4titude-Vision Plate™ 96-well microplates at 3 x $10^5$ cells/ml in 100 μl of RPMI 1640 medium and incubated overnight at 37˚C, 5% $CO_2$ and a humidified atmosphere. A further 50 μl of fresh growth medium was then added and the cells were treated in triplicate and incubated for 24 hours. MitoXpress® Xtra solution (10 μl) was then added to each well except the blank. All wells were gently overlaid with 100 μl of mineral oil. The plate was then inserted into Varioskan Flash multimode spectrophotometer (Thermofischer Scientific) and time-resolved fluorescence (TRF) was measured kinetically for 3 hours with an excitation wavelength of 380 nm and an emission wavelength at 650 nm and a delay time of $10\mu s$. Fluorescence signal intensity was plotted versus time and linear regression was applied to the linear portion of each curve to obtain the slopes. The slope obtained from a blank was subtracted from a test value to give a corrected fluorescence value.

## Lactate assay

After 48-hour treatments with either LPS and polymyxin B or a combination thereof, 10 μL of cell culture supernatant was carefully aspirated. A glycolysis reaction solution (6 ml) containing 60 μl each of glycolysis assay substrate (Cayman Chemicals, 600451) and co-factor (Cayman Chemicals, 600454) was made into 5.82 ml of assay buffer. L-lactate standards (10, 5, 2.5, 1.25, 0.625, 0.313, 0.156 and 0 mM) were prepared from the 10 mM stock solution (Cayman Chemicals, 600455) to generate a standard curve. 10 μl of each sample, in duplicate, were transferred into appropriate wells in a clear, flat-bottomed 96-well plate. Two wells were allocated for the blank, into which culture medium was transferred. An equal volume of L-lactate standards was also transferred into appropriate wells on the 96-well plate also in duplicate. 100 μl of the reaction solution was added using a multi-channel pipette to all wells. The plate was incubated at room temperature on an orbital shaker for 30 minutes. The absorbance was read at 490 nm using a Multiskan™ GO Microplate Spectrophotometer (Thermofischer Scientific, MA, USA). The average absorbance values were computed and the blank absorbance was subtracted from all values to obtain the corrected absorbance values. The corrected absorbance values for each of the standards were plotted against their corresponding concentration values to obtain a standard curve. The L-lactate concentrations for experimental samples were finally computed using the formula:

$$[L - lactate\ mM] = \left(\frac{Absorbance}{slope}\right) - y_{-intercept}$$

## Measurement of mitochondrial membrane potential

Following 48-hour treatments, cells were pelleted by centrifugation at 1500 rpm for 5 minutes using the Sigma 1–15 benchtop centrifuge. Positive control and negative control samples were FCCP (Sigma-C2920)—and 95% ethanol-treated samples respectively. The supernatant was discarded, and the pellet resuspended gently in 400 μl of JC-10 dye-loading solution (MAK160-1KT). The cells were incubated at 37˚C, 5% $CO_2$ and 95% humidity for 60 minutes. Fluorescence intensity was monitored using FL1 (green) and FL2 (orange) fluorescent channels in a BD Acuri C6 flow cytometer (BD Biosciences, CA, and USA). For proper gating, an untreated unstained sample was included to determine the location of non-fluorescent cells in the quadrants and the gates for all other samples were set accordingly (S1 Fig). Fluorochrome spill over between FL1 and FL2 channels was corrected by fluorescence compensation. FL1 was corrected by subtracting 3.2 units from FL2, while FL2 was corrected by subtracting 7.5 from FL1. All samples were run in triplicates. FCCP dissolved in 95% ethanol at 10 μM was added as a positive control. A vehicle-only control (cells treated with 95% ethanol only) was also included.

## Cell cycle analysis

After treatment with either LPS and Polymyxin B or their combination for 24 and 48 hours, the cells were fixed in 70% prechilled (at -20˚C) ethanol and then stored at -20˚C for 24 hours.

For propidium iodide staining, the cell suspensions were centrifuged using the Sigma 1–15 benchtop centrifuge at 3 000 rpm for 5 minutes to pellet the cells. The ethanol supernatant was discarded and the pellet washed twice by adding 1 ml of 1X PBS to cell pellet, resuspending and centrifuging for 5 minutes. 300 μl of FxCycle™PI/RNase (Molecular Probes, F10797) solution was then added to each pellet. Each tube was vortexed gently followed by incubation at room temperature in the dark for 30 minutes. Thereafter, samples were analysed using a BD FACS Aria III flow cytometer (BD Biosciences, CA, USA) with wavelength set at 488 nm. The

sample flow rate was set to 'slow', for proper interrogation at the flow cell. 10 000 events were acquired and the sample population was gated by identifying singlet populations on the forward scatter versus side scatter plot. PI intensity was read on an FL3 channel at a wavelength of 488 nm.

### Apoptosis and necrosis assay

Following cell cycle analysis, we observed a significant amount of sub G0/G1 population of cells (representing dead cells) after 48 hours of treatment of THP-1 cells with LPS decided to investigate the mode of cell death using an apoptosis/necrosis cell death assay. The Dead Cell Apoptosis Kit with Annexin V Alexa Fluor™ 488 & Propidium Iodide (PI) (Life Technologies-V13241) was used following the manufacturer's instructions.

### Pathway analysis

In order to investigate possible relationships between upregulated cytokines and enhanced glycolysis observed from increased lactate levels following LPS treatment, pathway analysis was done using GeneMANIA which predicts the function of genes of interest as well as possible protein-protein interaction networks. It uses weights that indicate the predictive value of each selected data set for the query. These weights are assigned to networks based on how significant they are to the query genes. Non-relevant networks have zero weightings, and are not shown in the results. The network weighting algorithms used by GeneMANIA have been validated in previous studies [24,25].

### Statistical analysis

Statistical analyses were done using a two-tailed Student's t-test on the GraphPad Prism webserver, comparing two groups at a time. Results were considered statistically significant at $p$ values <0.05. Unless otherwise stated, the data are presented as the mean ± standard deviation of three independent experiments.

## Results

### Activation of TLR-4 influences the metabolic programme in THP-1 cells

Having shown that LPS activation enhanced Toll-like receptor 4 (TLR4) expression in THP-1 cells (S1 Fig), we investigated whether LPS perturbed their metabolic programme by measuring glycolytic activity, oxidative phosphorylation and mitochondrial membrane potential (MMP) (Fig 1B–1E). TLR4 transcription was maximal at about 5 ng/ml and declined at the higher concentration of 20 ng/ml. Lactate measurements indicate that THP-1 cells already have a glycolytic phenotype which is enhanced in 48 hours after stimulation by LPS in a concentration-dependent manner and is impeded by polymyxin B (Fig 1B). In the first 24 hours, LPS does not perturb this glycolytic phenotype. When used alone, dichloroacetate (DCA), which sustains endogenous pyruvate into the Tricarboxylic Acid Cycle and methyl pyruvate, which represents exogenous pyruvate, reverse the glycolytic phenotype as shown by the reduced lactate and increased oxygen consumption (Fig 1C and 1D). Interestingly, the effects of DCA appear to be dependent on TLR4 activation because polymyxin B inhibits restores the glycolytic phenotype of DCA and LPS-treated cells. In contrast, exogenous pyruvate reverses the glycolytic phenotype by a different mechanism, as polymyxin B does not alter lactate production. However, when LPS and exogenous pyruvate appear to reverse the Warburg Effect as they reduce the observed glycolytic flux (Fig 1C) and increase OXPHOS (Fig 1D). This

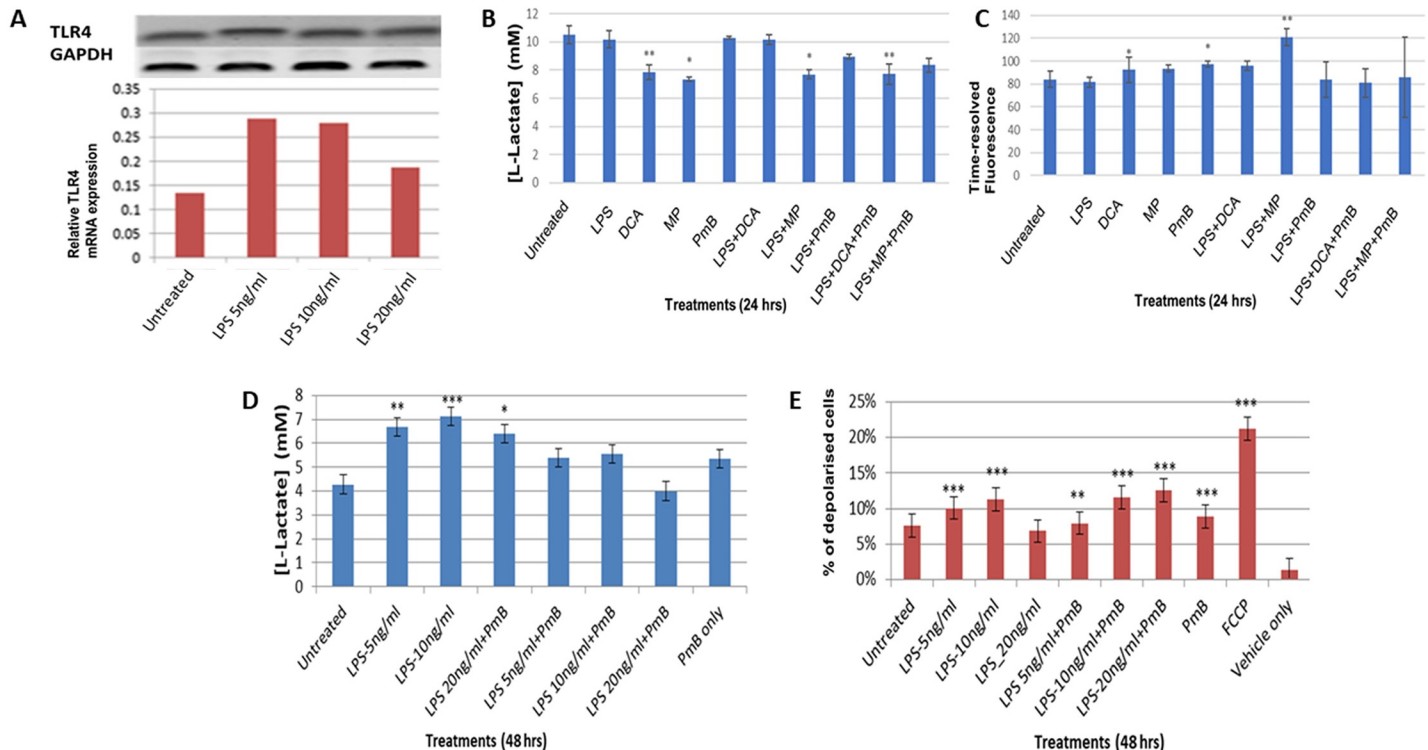

**Fig 1. LPS induces a dose-dependent increase in TLR4 mRNA expression and induces metabolic reprogramming in THP-1 cells. A** RT-PCR analysis following LPS treatments (5, 10 and 20 ng/ml) for 24 hours. Expression was normalised against GAPDH as reference gene. **B** L-lactate concentrations following 24-hour treatments with LPS and Polymyxin B (PmB). **C** changes in time-resolved fluorescence (TRF) following 24-hour treatments, indicative of oxygen consumption. **D** L-lactate production after 48-hour treatments with LPS and LPS-polymyxin B combination. **E** mitochondrial membrane depolarisation following treatments with LPS (5, 10 and 20 ng/ml) and co-treatments with PmB. Fluorescence intensity was monitored using FL1 and FL2 fluorescent channels in a BD Acuri C6 flow cytometer. A Student t-test was used to generate the p-values which compared the difference between the untreated and treated sample values. The data is represented as mean ± SD from 3 independent experiments (* indicates p <0.05, ** indicates p <0.01, *** indicates p <0.001). GraphPad Quick Calcs software was used to compute all statistics.

suggests that exogenous pyruvate is more effective in reversing the Warburg when compared to endogenous pyruvate.

Our data indicate that LPS induced a significant drop in MMP (P values ± SD = 0.0001 ± 0.013 and 0.0001 ± 0.014) following treatment with 5 and 10 ng/ml LPS respectively. Polymyxin B did not inhibit this LPS-mediated membrane depolarisation suggesting that it may be governed by a TRL4-independent mechanism. However, LPS caused these changes in mitochondrial membrane polarization in a concentration-dependent manner (Fig 1E, S1 Fig). The loss in mitochondrial membrane potential is often associated with apoptosis [26,27].

## LPS suppresses $G_2/M$ progression in THP-1 cells

We have previously shown that exogenous pyruvate kills breast, lung and ovarian cancer cell lines but promotes the survival of a normal fibroblast cell line. Furthermore, exogenous pyruvate protects the normal cell line from genotoxic stress caused by irinotecan [5]. Since monocytes are often exposed to stress caused by infectious agents, we investigated the response of THP-1 cells to LPS in order to understand the effect of superimposing infection on a cancerous monocyte. We also tested how THP-1 cells respond to exogenous pyruvate with or without LPS. The cells were treated as shown in (Fig 2A–2L) and the cell cycle progression and type of cell death were measured by flow cytometry following Annexin V staining to determine the

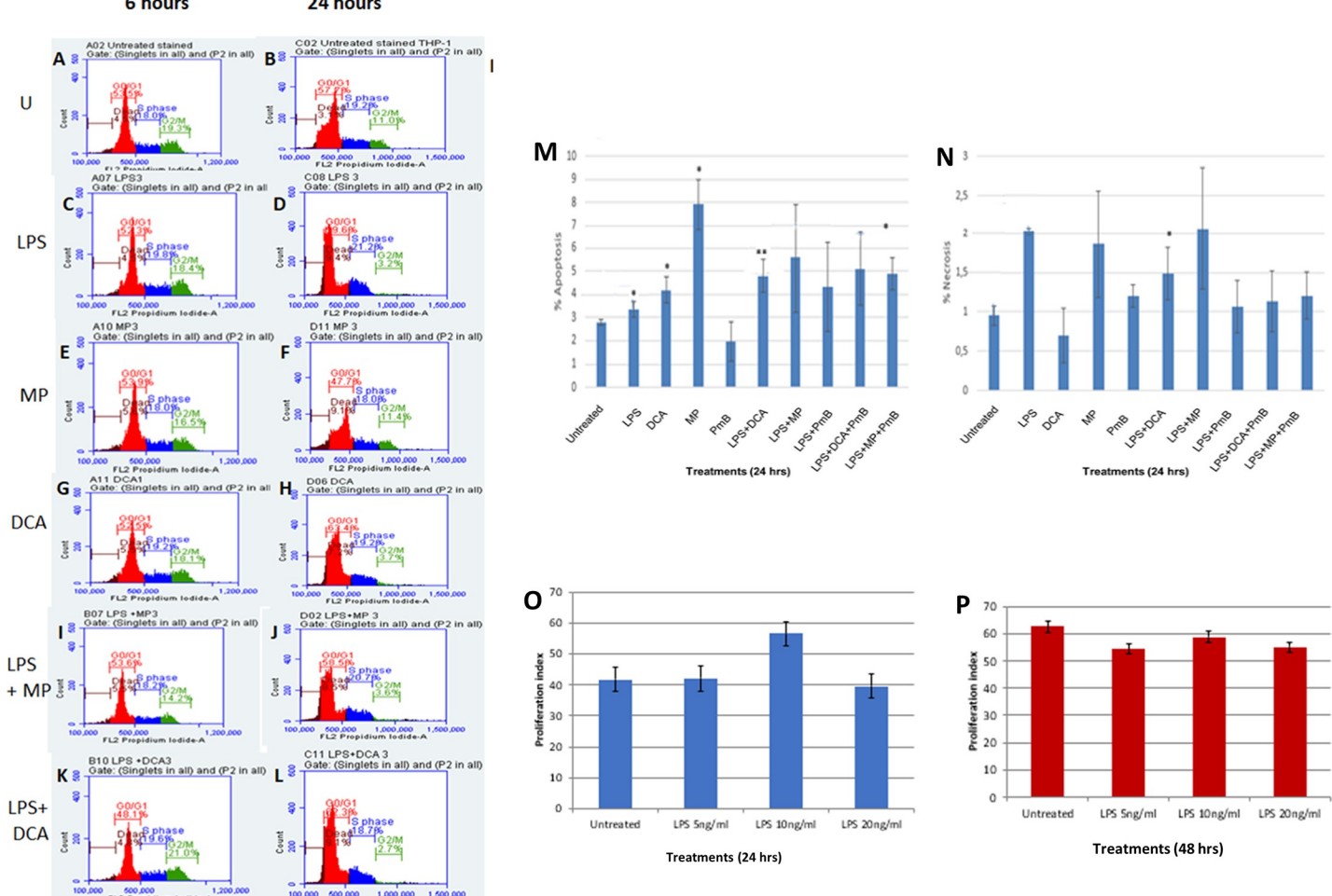

**Fig 2. The effects of LPS and perturbation of glucose metabolism on the cell cycle and cell viability. A to L** THP-1 cell cycle analysis in 6 and 24 hours. Cell were treated with permutations LPS (5 ng/ml), MP (0.08%) DCA (10 mM) as indicated. Cell death analysis shows proportion of cells undergoing apoptosis (**M**) or necrosis (**N**) after treatments as indicated for 24 hrs. Following cell cycle analysis, the Proliferation Index with 24hrs and 48 hrs (**O** and **P** respectively) was computed using the following formula: $[(G_2M+S)/(G_0G_1+S+G_2M)]X100$. A Student t-test was used to generate the p-values which compared the difference between the untreated and treated sample values. Data is represented as mean ± SD from 3 independent experiments (* indicates p <0.05, ** indicates p <0.01, *** indicates p <0.001). GraphPad Quick Calcs software was used to compute all statistics.

type of cell death. Overall, cells treated with LPS appeared to undergo S-phase cell cycle arrest. Detailed cell cycle changes are shown in (S2 Fig). In contrast to methyl pyruvate, DCA caused S-phase cell cycle arrest when measured at 24 hrs (Fig 2E–2H). Both DCA and methyl pyruvate did not reverse the LPS-induced S-phase arrest. (Fig 2I–2L). It is notable that unlike the non-myeloid cancer cell lines mentioned earlier, the reversal of the Warburg Effect using exogenous pyruvate does not kill the THP-1 cells.

Although the number of dead cells as observed by propidium iodide staining did not differ remarkably with all the treatments, Annexin V staining revealed that the proportion of cells killed by apoptosis and necrosis differed remarkably. LPS killed cells mainly by necrosis, which seems to be mediated by TLR-4 as it was reduced by polymyxin B. In contrast, exogenous pyruvate killed them by apoptosis. Moreover, methyl pyruvate did not affect LPS-induced necrosis although it reduced LPS-induced apoptosis (Fig 2M and 2N) and (S3 Fig). The proliferation index fluctuated over 48 hours, but it did not differ remarkably between treated and untreated cells (Fig 2O and 2P).

## LPS induces pro-inflammatory chemokines, IL-8, RANTES, MIPα, MIPβ and MDC

To further investigate the signalling pathway activated by LPS we treated the THP-1 cells with LPS (5 ng/ml) and polymyxin B (10 μg/ml) for 24 hours and measured downstream effector cytokine expression using a TLR-induced multi-analyte ELISA array kit. The array simultaneously profiled the expression of 12 inflammatory cytokines namely: TNF-α, IL1β, IL6, IL12, IL17A, IL8, MCP-1, RANTES, MIP-1α, MIP-1β, MDC and Eotaxin. We found that LPS induces IL-8, RANTES, MIPα, MIPβ and MDC that are pro-inflammatory chemokines (Fig 3A). Their induction was abrogated by polymyxin B. To validate the cytokine expression as shown by screening with the ELISA array, we performed RT-PCR on total RNA samples that were extracted from cells treated with LPS and LPS/Polymyxin B in biological duplicates for 24 and 48 hours. We used gene-specific primers for RANTES, IL-8, and MDC designed using the NCBI Primer blast software. This confirmed the LPS-induced transcription of genes that encode these cytokines in a concentration-dependent manner. Notably, known LPS-induced

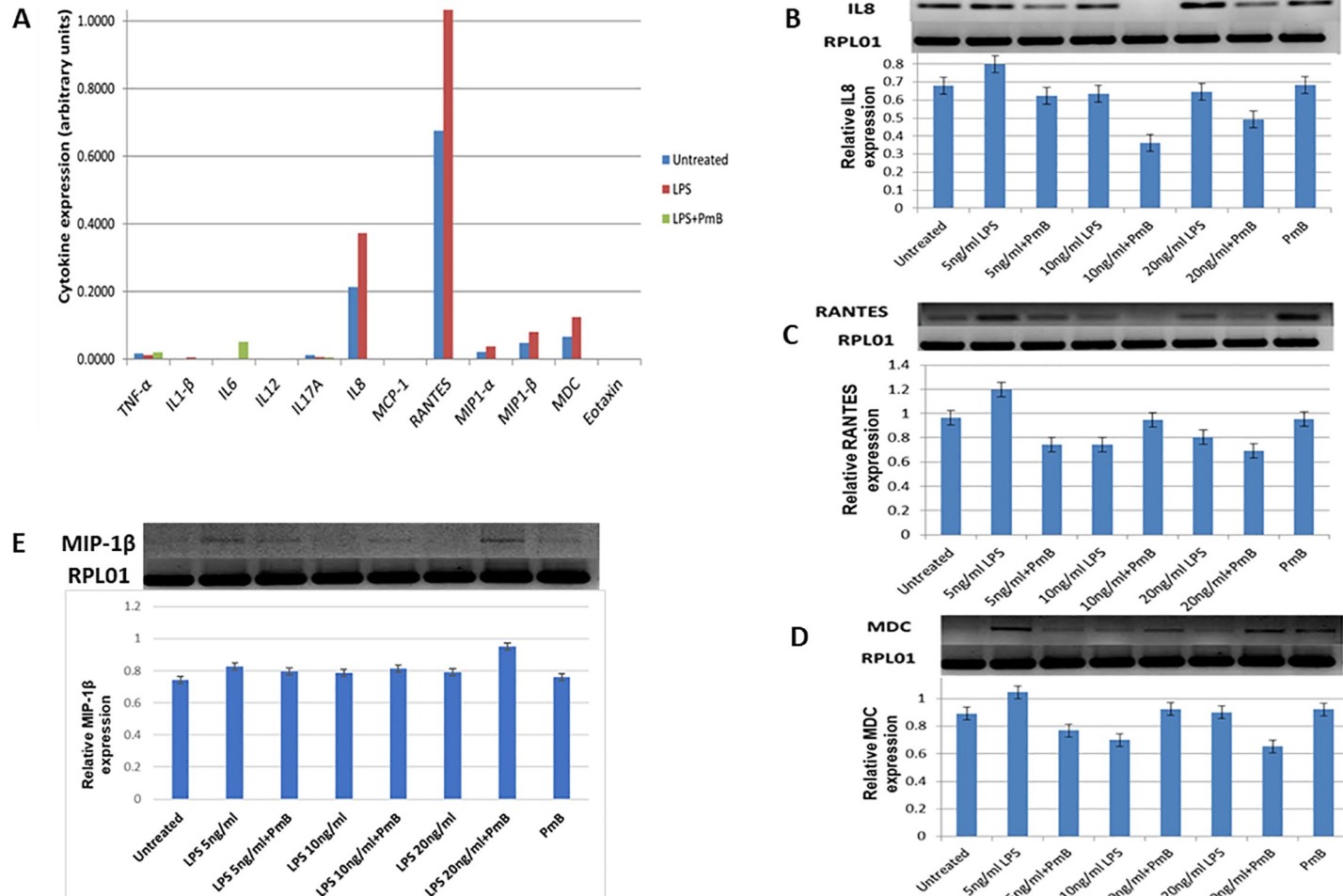

**Fig 3. Lipopolysaccharide-induced cytokine profile. A** shows induction of cytokines using an ELISA array for measuring 12 cytokines (shown on the x-axis) following treatment with LPS (5 ng/ml) and polymyxin B (PmB) (10 μg/ml) for 24 hours in a single detection experiment. The absorbance ($A_{570nm}$-$A_{450nm}$) data were obtained using a Multiskan Go microplate spectrophotometer. **B-E** show RT-PCR for IL8, RANTES, MDC and MIP-1β expression following 24-hour treatment with permutations of LPS and PmB as indicated. Each RT-PCR result is quantified by densitometry shown in the lower panels. The data were normalised with reference gene, RPL01. The data are represented as mean ± SD from two independent experiments.

cytokines such as TNF-α, IL-1β and IL-6 were not induced in our experiments (Fig 3B–3E) and (S4 Fig).

## Interaction pathway analysis suggests that the upregulated chemokines interact with glucose transporters and glycolytic pathway enzymes

To determine whether the affected cytokines have any functional impact on monocytes, we investigated the potential molecular interactions involving the four LPS-induced chemokines by *in silico* pathway analysis using GeneMANIA [24]. We also used the Integrative Multi-species Prediction (IMP) [28] to predict LPS-induced functional networks in THP-1 cells. Molecular analysis using GeneMANIA indicates that induced chemokines namely; RANTES, IL8, MIP-1β and MDC are co-expressed with their receptors, glucose transporters and with glycolysis enzymes (Fig 4A and Table 2).

More functional analysis using the IMP webserver reveals two distinct functional networks (Fig 4B). One set includes glycolysis enzymes as well as HSP90AB1 and YWHAZ (14-3-3ζ), both of which are associated with protein-folding and promote tumour formation and cancer cell proliferation [29]. The other group consists of anti-apoptotic molecules, cell adhesion molecules, glucose transporters and chemokine receptors. Notably, IL-8 and RANTES are the most upregulated by LPS-mediated TLR4 activation and both appear to dominate the interactions in this group. The anti-apoptotic proteins include B-cell lymphoma 2-related protein Z1 (BCL2A1) an NFκB target with strong pro-survival functions and Tumour Necrosis Factor-alpha protein 3 (TNFA1P3) which inhibits TNFα-mediated NFκB activation and TNF-mediated apoptosis [30,31]. In addition, this group comprises the classical glucose transporters SLC2A3 (GLUT3) which were found to mediate elevated glycolysis in colon cancer cells [32] and SLC2A14 (Glut14) which is a duplication of GLUT3 and is known to be specifically expressed in testis [33].

Finally, the software predicted interaction with the basic helix-loop-helix protein BHLHE40 which is integral to Vitamin D receptor-mediated signaling [34] and adrenomedullin (ADM) which is implicated in initiation and propagation of inflammatory response and known to be highly upregulated in LPS-treated macrophages [35]. The two group appear to be interlinked via the hypoxia-inducible factor (HIF-1), the prototypical regulator of oxygen homeostasis. Taken together, these observations suggest that the induced cytokines probably have an autocrine mechanism and may be involved in regulating metabolism and cell survival.

## Discussion

Although, cancer cells are often exposed to infectious environments, there are still unanswered questions about how they respond to infectious agents and commensals such as bacteria. A good example of this is cancers of the colon. We investigated the response of one representative of an innate immune cell to assess the cellular responses to infection by an already cancerous cell. We have shown that TLR-4 signalling can mediate the glycolytic phenotype in THP-1 cells enhancing it when they are activated by LPS. This is accompanied by upregulation of a subset of pro-inflammatory chemokines. Furthermore, LPS appears to have an impact on cell cycle progression as THP-1 cells undergo S-phase cell cycle arrest within 24 hours after treatment. Moreover, there is minimal cell death, mainly by necrosis, in LPS-treated THP-1 cells. In contrast to non-immune cancer cell lines, perturbation of this glycolytic programme by exogenous pyruvate does not kill the THP-1 cells. Instead, cell division appears to proceed in the same manner as untreated cells. This suggests that in monocytes, modulation of the Warburg Effect may be governed by a different molecular mechanism. Alternatively, it suggests that the impact of the Warburg Effect is context-dependent.

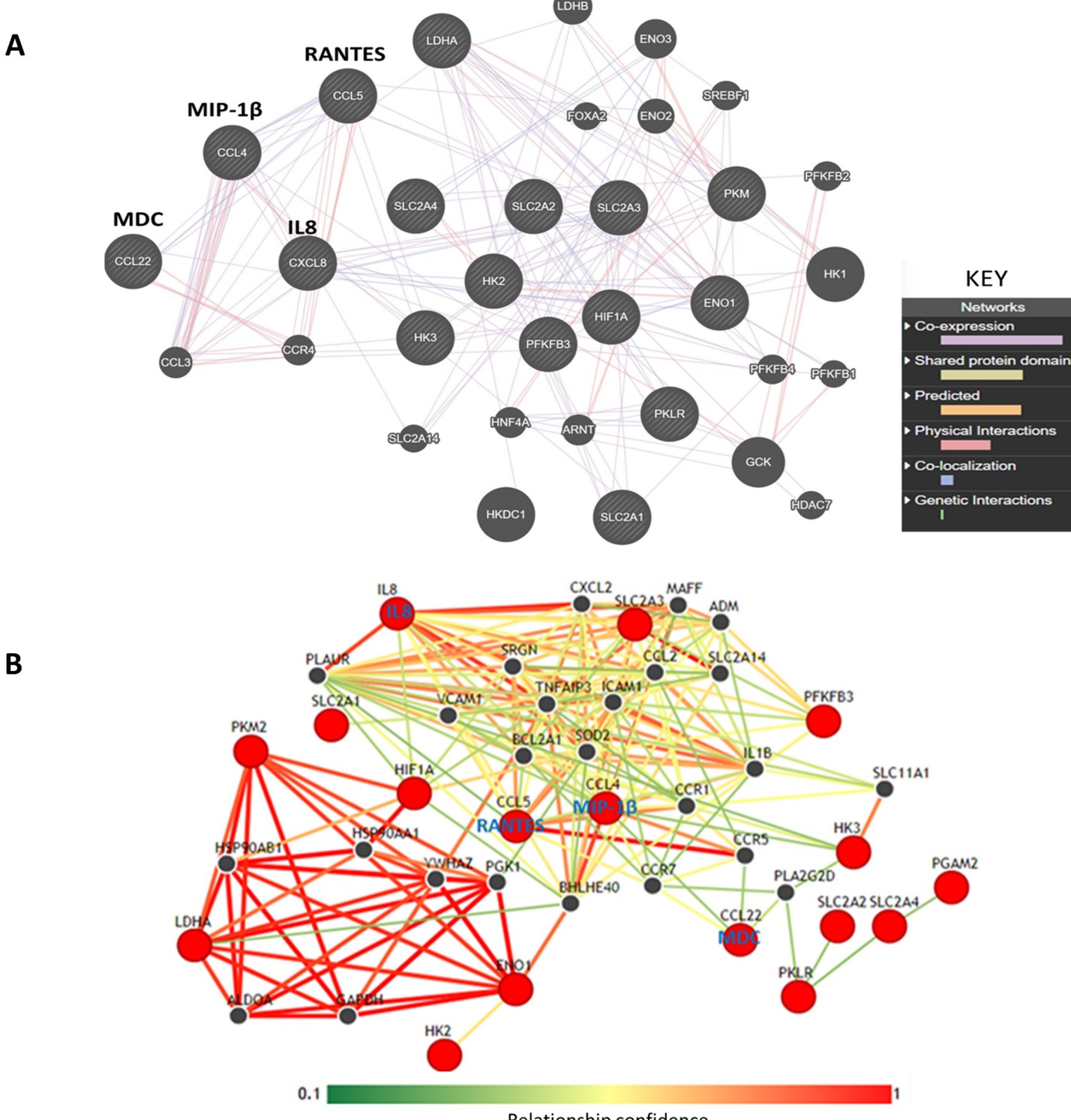

**Fig 4. Interleukin 8, RANTES, MIP-1 beta and MDC show co-expression and other potential interactions with glycolytic enzymes.** The diagram shows gene interaction networks based on the upregulated cytokines. The interaction networks were generated using upregulated cytokines and glycolytic genes as input into the GeneMANIA and IMP webservers. **A** and **B** show interactions generated by GeneMANIA and IMP respectively. In **A**, the nodes represent genes while the edges represent interactions. All striped nodes show input genes while unstriped nodes represent other interacting genes. The predicted type of interaction is colour coded (see the inserted key). Interactions were considered statistically significant when an FDR (False discovery rate) value is < 0.05. FDR<0.05 represents a 5% chance of getting a false statistically significant result. GeneMANIA computes FDR values using the Benjamini-Hochberg procedure, which is built in the webserver. **B** shows predicted

interaction network of the cytokines and glycolytic proteins using the IMP webserver. The relationship confidence bar shown on the figure gives the strength of the interaction based on network weights. This means that red edges represent stronger interaction confidence while green interactions show a weaker confidence. The red nodes represent input molecules while the black nodes represent other interacting proteins in the network.

We have noted that known LPS-induced cytokines such as TNF-α, IL-1β, and IL-6 are not induced in our study. In previous studies, it has been shown consistently, that IL-1β, TNF-α. IL-6, IL-8 and IL-10 are stimulated by *E. coli* LPS in THP-I cells [13,36]. However, in our study, IL-1β, TNF-α and IL-6 are not affected. One explanation could be that these studies used much higher concentrations (μg) compared to the nanogram amounts used here. As a screening exercise, this experiment was not repeated but the upregulated cytokines were further measured by PCR whose results were consistent with ELISA in two biological replicates. It is known that LPS can induce NF–B activated gene expression using a different pathway to TNFα because a dominant negative TRAF2 mutant does not inhibit it. It was also shown that LPS-induced activation in THP-1 cells does not involve the Interleukin 1 receptor (IL-1R) because gene expression is not inhibited by IL-1R antagonists although dominant negative mutants of molecules upstream of IKK/NFκB point inhibited LPS-induced NFκB activation. Thus, TLR4 was identified as the mediator of another pathway that converged at IKK/NFκB [37].

In addition to modulating the transcription of TLR4 and pro-inflammatory chemokines, we have found that LPS enhances the glycolytic phenotype of THP-1 cells and causes S-phase cell cycle arrest after 24 hours ostensibly by inhibiting G2/M progression. The mechanism by which LPS induces cell cycle arrest is still unclear. In murine bone marrow-derived macrophages (BMM), LPS was shown to arrest the cell cycle at $G_1$[38] [39]. Although LPS induces the mutagenic free radical nitric oxide (NO) via the nitric oxide synthase 11 gene [40], it is doubtful that NO is responsible for cell cycle arrest in macrophages. Firstly, a study designed to investigate this matter showed that LPS inhibits proliferation of NOS$^{-/-}$ murine BMM. Secondly, LPS-induced cell cycle arrest in wild type BMM was not reversed by a NOSII inhibitor

**Table 2. Genes used for *GeneMANIA* and IMP pathway analysis.**

| NCBI Gene ID | Gene | Common name |
|---|---|---|
| **Upregulated cytokines** | | |
| 3576 | CXCL8 | Interleukin 8 (IL-8) |
| 6352 | CCL5 | Regulated Upon Activation Normal T cell Express Sequence (RANTES) |
| 6351 | CCL4 | Macrophage inflammatory protein-1 beta (MIP-1β) |
| 6367 | CCL22 | Macrophage-Derived Chemokine (MDC) |
| **Glycolytic genes investigated** | | |
| 3099 | HK2 | Hexokinase, isoform 2 |
| 3101 | HK3 | Hexokinase, isoform 3 |
| 3939 | LDHA | Lactate dehydrogenase A |
| 5224 | PGAM2 | Phosphoglycerate mutase, isoform 2 |
| 6513 | SLC2A1 | Glucose transporter 1 (GLUT1) |
| 6517 | SLC2A4 | Glucose transporter 4 (GLUT4) |
| 2023 | ENO1 | Enolase 1 |
| 5163 | PDK1 | Pyruvate dehydrogenase kinase1 |
| 5315 | PKM | Pyruvate kinase |
| 3091 | HIF-1α | Hypoxia-inducible factor 1 alpha |
| 5209 | PFKFB3 | 6-phosphofructo-2-kinase/fructose-2,6-biphosphatase 3 |

[38]. In the present study, THP-1 cells, which represent undifferentiated macrophages, are shown to accumulate at the S-phase when exposed to LPS. It has been shown that inhibition of cyclin A, cyclin E, as well as their kinase CDK-2, causes cells to accumulate at S-phase [41]. However, we have found no reports of LPS inhibiting these checkpoint molecules.

The results as shown here resemble those produced by LPS in mouse bone marrow-derived macrophages whereby it inhibited proliferation of NOSII$^{-/-}$ mouse macrophages indicating that LPS can promote cell proliferation in the absence of nitric oxide. However, unlike the THP-1 cells, these primary macrophages underwent G1 cell cycle arrest [38].

Surprisingly, we found that the exogenous pyruvate promotes cell cycle progression in THP-1 cells in contrast to non-myeloid cell lines which were previously shown to be sensitive to reversal of the Warburg Effect by exogenous pyruvate [5,42]. Although methyl pyruvate promotes cell cycle progression in a normal cell line [5] and in THP-1 cells as shown here, it does not unlock the G2/M cell arrest of LPS-treated THP1 cells. Moreover, enhancing endogenous pyruvate by using DCA does not alter the $G_2$/M-phase cell cycle arrest caused by LPS. This suggests that subverting the glycolytic metabolic programme in the myeloid cell line does not activate apoptotic pathways as it does in the non-myeloid cancer cell lines.

Upon activation by LPS, the expression of pro-inflammatory chemokines was induced in THP-1 cells. This is interesting because myeloid lineage cells play a key role in polarizing innate and adaptive immune response through their differential chemokine expression [43]. The significance of the set of induced chemokines needs some evaluation. RANTES and MIP are pro-inflammatory chemokines which are ligands for the CCR5 receptor found on monocytes, macrophages, Th1 cells, activated T cells and on natural killer (NK) cells suggesting that activation of monocytes by LPS would influence these cell types. The MDC chemokine, which is involved in both pro-inflammatory and homeostatic activities is a ligand for CCR4 and is expressed on immature dendritic cells, monocytes, Th2 cells, natural killer and T skin cells. This profile then shows that LPS activation stimulates expression of 3 chemokine effectors with autocrine function. Expression of Interleukin 8, which binds to CXCR1 and CXCR2 receptors on neutrophils, is also enhanced.

It is worth considering if this chemokine profile is related to the enhancement of the glycolytic phenotype in LPS-induced THP-1 cells. The pathway prediction study presented here suggests that LPS induces a chemokine profile that is connected to anti-apoptotic signals, cell adhesion molecules, glucose transporters and many glycolysis enzymes. BLC2A1, in particular, is expressed in haematopoietic cells to facilitate the survival of a subset of leukocytes inflammation [44]. Interestingly the hypoxia-inducible factor is strongly connected to the glycolysis proteins and to many others in the second group such as adrenomedullin and the glucose transporters which are known HIF-1 targets. Although it has been reported that LPS does not induce HIF-1 in undifferentiated THP-1 cells [23], it is a major role player on oxygen homeostasis and on cell survival under hypoxic conditions [45].

## Conclusions

The data presented here suggest that LPS activation of THP-1 cells enhances their glycolytic phenotype and induces a set of proinflammatory chemokines. LPS also appears to induce S-phase cell cycle arrest at least 6 hours after treatment and minimal cell death even after 24 hours. The reversal of the glycolytic phenotype by exogenous pyruvate does not affect the cells as expected. This raises some complexity about the Warburg Effect in myeloid cells. Altogether, the molecular analysis conducted here suggests that the chemokine induction is linked to glycolysis and cell survival of the LPS-activated monocytic leukaemia cell line but not to

proliferation. In future studies, it is necessary to test the impact of this molecular profile on the cell cycle because the mechanism by which the observed cellular dynamics occur is not clear.

## Supporting information

**S1 Fig. LPS upregulates the mRNA expression of IL8, RANTES and MDC, antagonised by PmB. A** and **D** show RT-PCR gel images and corresponding densitometry results for IL8**, B** and **E:** RANTES and **C** and **F:** MDC, following 24-, and 48-hour treatments respectively, with 5, 10 and 20 ng/ml of LPS and co-treatments with 10 μg/ml PmB. All genes were normalised with reference, RPL01.
(TIF)

**S2 Fig. Time-dependent effects of various drugs on the cell cycle progression in THP-1 cells.** The cells were treated with 5 ng/ml LPS (Lipopolysaccharides), 10 mM DCA (dichloroacetate), 0.08% MP (Methyl pyruvate) and/or 10 μg/ml PmB (Polymyxin B) for 6, 12,18, and 24 hours. **Sub G0/G1** phase is shown in **maroon**; **G0/G1** phase in **red**, **S-phase** in blue, and **G2/M phase** in **green**. The cell cycle assay was performed using BD Acurri $^{TM}$ flow cytometer. The data represented here is a representative of three separate experiments. Florescence data were acquired on the FL2 (orange fluorescence) channel. **B. Cell cycle analysis in THP-1 cells.** Cells were treated with 5 ng/ml of LPS, 0.08 MP (methyl pyruvate), 10 mM DCA (dichloroacetate) and/or 10 μg/ml PmB for 6, 12, 18 and 24 hours **(i)**–**(iv)**. The **blue bars** represent sub-G0/G1 cell populations; **orange bars** show G0/G1 cells, **grey bars** indicate S-phase populations and **yellow bars** depict G2/M cell populations. The data are represented as mean ± SD from 3 independent experiments (* indicates $p < 0.05$, ** indicates $p < 0.01$, *** indicates $p < 0.001$. All statistics were computed using GraphPad Quick Calcs software.
(JPG)

**S3 Fig. Effects of LPS, MP and DCA on cell viability in THP-1 cells following 24 hours of treatment.** Each diagram represents a treatment. Annexin V/PI stained THP-1 cells following treatment with either 5 ng/ml LPS, 0.08% MP, 10mM DCA,10 μg/ml PmB and combination of these treatments in comparison with untreated cells for 24 hours. Each quadrant represents populations of viable (lower left), early apoptotic (lower right), late apoptotic (upper right) and necrotic (upper left) cells The data were acquired using a BD Acuri C6 flow cytometer with propidium iodide (PI) fluorescence monitored on the FL3 (red fluorescence) channel (shown on the y-axis) while annexin V-alexa 488 of the FL1 (green fluorescence) channel (shown on the x-axis).
(TIF)

**S4 Fig. LPS induces mitochondrial membrane depolarization; independent of polymyxin B. A** Each diagram is a representative of three independent treatments. The cells were treated with 5, 10 and 20 ng/ml LPS and/or polymyxin B for 48 hours. The x-axis represents the FL2 (Green fluorescence) channel, while the y-axis shows FL2 (orange fluorescence) channel. The **lower left quadrant** shows unstained cells, **lower right quadrant**: green fluorescent (depolarised) cells; and the **upper right quadrant**: orange fluorescent (polarised) cells. FCCP (Carbonyl cyanide-*4*-(trifluoromethoxy) phenylhydrazone), dissolved in 95% ethanol was used as positive control. 95% ethanol was used as the vehicle control. **B.** MMP analysis showing significant membrane depolarisation following LPS treatment. The histogram shows mitochondrial membrane depolarisation following treatments with LPS (5, 10 and 20 ng/ml) and co-treatments with polymyxin B (PmB). Fluorescence intensity to assess mitochondrial membrane potential was monitored using FL1 (green) and FL2 (orange) fluorescent channels in a BD Acuri C6 flow cytometer. The **blue bars** show polarised cells while the **red bars** show the

percentage of depolarised cells. A Student t-test was used to generate the p-values which compared the difference between the untreated and treated sample values. The data is represented as mean ± SD from 3 independent experiments (* indicates $p < 0.05$, ** indicates $p < 0.01$, *** indicates $p < 0.001$). GraphPad Quick Calcs software was used to compute all statistics. (JPG)

## Acknowledgments

We would like to thank Mr Garland More for technical assistance on the Oxygen Consumption Assays.

## Author Contributions

**Conceptualization:** Monde Ntwasa.

**Data curation:** Philemon Ubanako, Ntombikayise Xelwa.

**Formal analysis:** Philemon Ubanako, Monde Ntwasa.

**Funding acquisition:** Monde Ntwasa.

**Investigation:** Philemon Ubanako, Monde Ntwasa.

**Methodology:** Philemon Ubanako, Ntombikayise Xelwa, Monde Ntwasa.

**Supervision:** Monde Ntwasa.

**Writing – original draft:** Monde Ntwasa.

**Writing – review & editing:** Philemon Ubanako, Ntombikayise Xelwa, Monde Ntwasa.

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
