## [Decision Letter · Decision Letter 0]

22 Jul 2019

PONE-D-19-17316

LPS induces inflammatory chemokines via TLR-4 signalling and enhances the Warburg Effect in THP-1 cells

PLOS ONE

Dear Professor Ntwasa,

Thank you for submitting your manuscript to PLOS ONE. After careful consideration, we feel that it has merit but does not fully meet PLOS ONE’s publication criteria as it currently stands. Therefore, we invite you to submit a revised version of the manuscript that addresses the points raised during the review process.

WE received positive feed back from reviewers. However, some major issues were raised by one reviewer and require your attention.

We would appreciate receiving your revised manuscript by Sep 05 2019 11:59PM. To enhance the reproducibility of your results, we recommend that if applicable you deposit your laboratory protocols in protocols.io, where a protocol can be assigned its own identifier (DOI) such that it can be cited independently in the future. For instructions see: http://journals.plos.org/plosone/s/submission-guidelines#loc-laboratory-protocols

We look forward to receiving your revised manuscript.

Kind regards,

Partha Mukhopadhyay, Ph.D.

Academic Editor

PLOS ONE

Journal Requirements:

Reviewers' comments:

Reviewer's Responses to Questions

**Comments to the Author**

1. Is the manuscript technically sound, and do the data support the conclusions?

Reviewer #1: Yes

Reviewer #2: Yes

2. Has the statistical analysis been performed appropriately and rigorously? 

Reviewer #1: Yes

Reviewer #2: Yes

3. Have the authors made all data underlying the findings in their manuscript fully available?

Reviewer #1: Yes

Reviewer #2: Yes

4. Is the manuscript presented in an intelligible fashion and written in standard English?

Reviewer #1: Yes

Reviewer #2: Yes

5. Review Comments to the Author

Reviewer #1: The authors have tried to address the role of infection on induction of metabolic program in THP-1 cells. Addressed that metabolic changes for these cells does not affect its viability and Warburg effect does not play a role. Interestingly, very few cytokines were expressed upon infection in these cells. I would like the authors to address the following concerns.

Comments:

1. I assume fig. 1a is the western blot for TLR protein. It does not seem upon induction with LPS 5ng/ml the protein does not increased while mRNA increased. How many repeats were carried out for western blot and if possible quantify after three repeats, if done.

2. No mention of western blot protocol in material method section. Authors has to be careful in this regard and update it.

3. Fig. 3a- I am not sure how many repeats were carried out for ELISA.

4. Fig. 3b-e – Again for western blots- I would like to see how many repeats were carried out and if possible can they quantify the western blots that can be comparable with the RT-PCR result.

5. This is because it is confusing for all the Figures in 3, as an example – the difference between mRNA to protein in Fig. 3b- compared to 5ng/ml+PmB to 10ng/ml LPS the mRNA level is same between these samples while the 10ng/ml LPS the protein level is high.

6. Similarly, for RANTES (Fig. 3C)- as by visualization 10ng/ml LPS has protein while 10ng/ml LPS with PmB no protein while the mRNA profile is different between these samples.

7. Thus, it is important to address this issue upon repeating western for all these genes (Fig. 3c to e) and quantify whether there is any mRNA and protein difference.

8. Since the authors have addressed the impact of cytokines and its linkage with downstream pathways. Why not address few genes that are linked to IL8 or RANTES by knocking down these genes to show its effect on its interactome?

9. I would also like to know NF kappa B expression upon these treats because it will be an important marker to show that even though NF kappa B expression increased in LPS treatment, only few of its target cytokines expression is altered.

Reviewer #2: In this manuscript, the authors analyzed cytokine production and Warburg effect induced by LPS in THP-1 AML cell line. While it is already widely known that LPS induces inflammatory cytokine production, the authors report an interesting finding in the glycolytic changes in THP-1 cells. The techniques used by the authors are sound, and the results fully demonstrate their conclusion.

6. PLOS authors have the option to publish the peer review history of their article (what does this mean?). If published, this will include your full peer review and any attached files.

Reviewer #1: No

Reviewer #2: No

---

## [Author Response · Author response to Decision Letter 0]

27 Aug 2019

Reviewer #1: 

1. I assume fig. 1a is the western blot for TLR protein. It does not seem upon induction with LPS 5ng/ml the protein does not increased while mRNA increased. How many repeats were carried out for western blot and if possible quantify after three repeats, if done.

Fig 1a is an RT-PCR showing TLR4 mRNA expression and not a Western blot. The reviewer must have made a mistake because the legend clearly indicates this. We have highlighted the section in Figure 1 legend that shows that the blots depict RT-PCR. No repeats were performed for TLR4 expression since it is well-known that THP-1 cells express TLR4. This was just to confirm expression and induction by LPS at the low concentrations compared to published ones. The error bars were generated accidentally because the “error bars” item was ticked off on the spreadsheet “Chart Elements” menu. We apologise for this. We generated a new graph and presented it in Fig 1a.

2. No mention of Western blot protocol in material method section. Authors has to be careful in this regard and update it.

 We did not perform any Western blot experiments. Hence, no Western blot protocol was included.

3. Fig. 3a- I am not sure how many repeats were carried out for ELISA

 The ELISA assay was performed in a single experiment as a screening tool as stated in the results and discussion section of the manuscript. We validated this expression profile by using RT-PCR in two biological replicates. We are convinced about the validity of the Array result because the RT-PCR expression profile is consistent with the Array. 

4. Fig. 3b-e – Again for western blots- I would like to see how many repeats were carried out and if possible can they quantify the western blots that can be comparable with the RT-PCR result.

No Western analyses were conducted in this submission. The legend of Figure 3 indicates that there were two biological replicates (highlighted).

5. This is because it is confusing for all the Figures in 3, as an example – the difference between mRNA to protein in Fig. 3b- compared to 5ng/ml+PmB to 10ng/ml LPS the mRNA level is same between these samples while the 10ng/ml LPS the protein level is high.

No Western analyses were conducted in this submission. The source of the confusion could be the lack of explanation for the bar graphs. We have now included in the legend that the lower panels represent quantification of the RT-PCR gels bands. They are not protein measurements.

6. Similarly, for RANTES (Fig. 3C)- as by visualization 10ng/ml LPS has protein while 10ng/ml LPS with PmB no protein while the mRNA profile is different between these samples.

 No Western analyses were conducted in this submission. Please see (5) above.

7. Thus, it is important to address this issue upon repeating western for all these genes (Fig. 3c to e) and quantify whether there is any mRNA and protein difference.

No Western analyses were conducted in this submission. Please see (5) above.

8. Since the authors have addressed the impact of cytokines and its linkage with downstream pathways. Why not address few genes that are linked to IL8 or RANTES by knocking down these genes to show its effect on its interactome?

We do agree that this study reveals a broad view and complexity of the molecular pathways induced by infection in the tumour microenvironment. However, at this stage the primary questions were about nature of the Warburg Effect in monocytes. The finer studies suggested by the reviewer are being considered for new studies where the tumour microenvironment is simulated in 3D cultures. We do appreciate these comments because they reinforce our views about the future experiments. 

9. I would also like to know NF kappa B expression upon these treats because it will be an important marker to show that even though NF kappa B expression increased in LPS treatment, only few of its target cytokines expression is altered.

NF Kappa B regulates multiple aspects of innate immunity and has two broad mechanisms: the (i) canonical and (ii) non-canonical pathways. It is not too surprising that not all effectors known to be mediated by it are induced as its activity is contextual (temporal and cell specific). It is, however, notable that in the profile we present, the chemokines (known to be NF-Kappa B targets) are induced but the cytokines (some known known to be NF Kappa B targets) are poorly induced or not at all). We have addressed the NF kappa B influence in the discussion section. Although we appreciate the interest of the reviewer in NF kappa B expression, we do not think that a focussed study on NF kappa be would be useful in understanding the Warburg Effect at this stage. It could probably enlighten us about cytokine expression pathways – which is not the primary interest of the manuscript.

Reviewer #2: 

In this manuscript, the authors analyzed cytokine production and Warburg effect induced by LPS in THP-1 AML cell line. While it is already widely known that LPS induces inflammatory cytokine production, the authors report an interesting finding in the glycolytic changes in THP-1 cells. The techniques used by the authors are sound, and the results fully demonstrate their conclusion.

 We are grateful for these comments.

---

## [Decision Letter · Decision Letter 1]

4 Sep 2019

[EXSCINDED]

LPS induces inflammatory chemokines via TLR-4 signalling and enhances the Warburg Effect in THP-1 cells

PONE-D-19-17316R1

Dear Dr. Ntwasa,

We are pleased to inform you that your manuscript has been judged scientifically suitable for publication and will be formally accepted for publication once it complies with all outstanding technical requirements.

With kind regards,

Partha Mukhopadhyay, Ph.D.

Section Editor

PLOS ONE

Additional Editor Comments (optional):

Reviewers' comments:

Reviewer's Responses to Questions

**Comments to the Author**

1. If the authors have adequately addressed your comments raised in a previous round of review and you feel that this manuscript is now acceptable for publication, you may indicate that here to bypass the “Comments to the Author” section, enter your conflict of interest statement in the “Confidential to Editor” section, and submit your "Accept" recommendation.

Reviewer #1: All comments have been addressed

2. Is the manuscript technically sound, and do the data support the conclusions?

Reviewer #1: Yes

3. Has the statistical analysis been performed appropriately and rigorously? 

Reviewer #1: Yes

4. Have the authors made all data underlying the findings in their manuscript fully available?

Reviewer #1: Yes

5. Is the manuscript presented in an intelligible fashion and written in standard English?

Reviewer #1: Yes

6. Review Comments to the Author

Reviewer #1: (No Response)

7. PLOS authors have the option to publish the peer review history of their article (what does this mean?). If published, this will include your full peer review and any attached files.

Reviewer #1: No

---

## [Editor Report · Acceptance letter]

12 Sep 2019

PONE-D-19-17316R1 

LPS induces inflammatory chemokines via TLR-4 signalling and enhances the Warburg Effect in THP-1 cells 

Dear Dr. Ntwasa:

I am pleased to inform you that your manuscript has been deemed suitable for publication in PLOS ONE. Congratulations! Your manuscript is now with our production department. 

With kind regards,

on behalf of

Dr. Partha Mukhopadhyay 

Section Editor

PLOS ONE